# A Closer Look at the Calibration of Differentially Private Learners

## Abstract

We systematically study the calibration of classifiers trained with differentially private stochastic gradient descent (DP-SGD) and observe miscalibration across a wide range of vision and language tasks. Our analysis identifies per-example gradient clipping in DP-SGD as a major cause of miscalibration, and we show that existing approaches for improving calibration with differential privacy only provide marginal improvements in calibration error while occasionally causing large degradations in accuracy. As a solution, we show that differentially private variants of post-processing calibration methods such as temperature scaling and Platt scaling are surprisingly effective and have negligible utility cost to the overall model. Across 7 tasks, temperature scaling and Platt scaling with DP-SGD result in an average 3.1-fold reduction in the in-domain expected calibration error and only incur at most a minor percent drop in accuracy.

## 1 Introduction

Modern deep learning models tend to memorize their training data in order to generalize better (Zhang et al., 2021; Feldman, 2020), posing great privacy challenges in the form of training data leakage or membership inference attacks (Shokri et al., 2017; Hayes et al., 2017; Carlini et al., 2021). To address these concerns, differential privacy (DP) has become a popular paradigm for providing rigorous privacy guarantees when performing data analysis and statistical modeling based on private data. In practice, a commonly used DP algorithm to train machine learning (ML) models is DP-SGD (Abadi et al., 2016). The algorithm involves clipping per-example gradients and injecting noises into parameter updates during the optimization process.

Despite that DP-SGD can give strong privacy guarantees, prior works have identified that this privacy comes at a cost of other aspects of trustworthy ML, such as degrading accuracy and causing disparate impact (Bagdasaryan et al., 2019; Feldman, 2020; Sanyal et al., 2022). These tradeoffs pose a challenge for privacy-preserving ML, as it forces practitioners to make difficult decisions on how to weigh privacy against other key aspects of trustworthiness. In this work, we expand the study of privacy-related tradeoffs by characterizing and proposing mitigations for the *privacy-calibration* tradeoff. The tradeoff is significant as accessing model uncertainty is important for deploying models in safety-critical scenarios like healthcare and law where explainability (Cosmides & Tooby, 1996) and risk control (Van Calster et al., 2019) are needed in addition to privacy (Knolle et al., 2021).

The existence of such a tradeoff may be surprising, as we might expect differentially private training to *improve* calibration by preventing models from memorizing training examples and promoting generalization (Dwork et al., 2015; Bassily et al., 2016; Kulynych et al., 2022). Moreover, training with modern pre-trained architectures show a strong positive correlation between calibration and classification error (Minderer et al., 2021) and using differentially private training based on pre-trained models are increasingly performant (Tramer & Boneh, 2021; Li et al., 2022b; De et al., 2022). However, we find that DP training has the surprising effect of consistently producing over-confident prediction scores in practice (Bu et al., 2021). We show an example of this phenomenon in a simple 2D logistic regression problem (Fig. 1). We find a polarization phenomenon, where the DP-trained model achieves similar accuracy to its non-private counterpart, but its confidences are clustered around either 0 or 1. As we will see later, the polarization insight conveyed by this motivating example transfers to more realistic settings.

Our first contribution quantifies existing privacy-calibration tradeoffs for state-of-the-art models that leverage DP training and pre-trained backbones such as RoBERTa (Liu et al., 2019b) and vision transformers (ViT) (Dosovitskiy et al., 2020). Although there have been some studies of miscalibration for differentially private learning (Bu et al., 2021; Knolle et al., 2021), they focus on simple tasks (e.g., MNIST, SNLI) with relatively small neural networks trained from scratch. Our work shows that miscalibration problems persist even for state-of-the-art private models with accuracies approaching or matching their non-private counterparts. Through controlled experiments, we show that these calibration errors are unlikely solely due to the regularization effects of DP-SGD, and are more likely caused by the per-example gradient clipping operation in DP-SGD.

Our second contribution shows that the privacy-calibration tradeoff can be easily addressed through differentially private variants of temperature scaling (DP-TS) and Platt scaling (DP-PS). To enable these modifications, we provide a simple privacy accounting analysis, proving that DP-SGD based re-calibration on a held-out split does not incur additional privacy costs. Through extensive experiments, we show that DP-TS and DP-PS effectively prevent DP-trained models from being overconfident and give a 3.1-fold reduction in in-domain calibration error on average, substantially outperforming more complex interventions that have been claimed to improve calibration (Bu et al., 2021; Knolle et al., 2021).

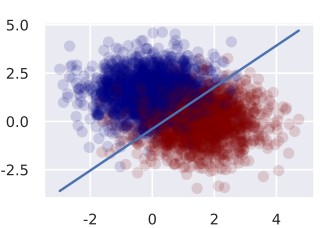 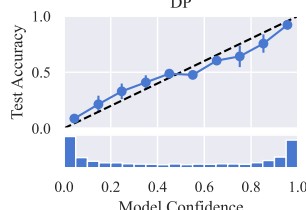 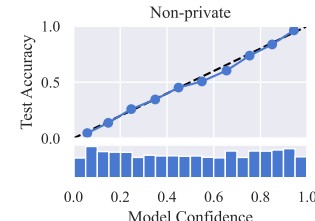

(a) Non-separable Gaussian Data     (b) Calibration comparison of logistic regression w and w/o DP

Figure 1: **DP-SGD gives rise to miscalibration for logistic regression.** (a) Logistic Regression model (blue line) with $\epsilon = 8$ on Gaussian data $\{(x_i, y_i)\}_{i=1}^n$ where $(x, y) \in \mathbb{R}^p \times \{1, -1\}$, $(x-b)|y \sim \mathcal{N}(0, I_{2\times 2})$, $b = (1.5, 0)$ if $y = 1$ else $b = (0, 1.5)$, and $y$ is Rademacher distributed. (b) Reliability diagram and confidence histogram. DP-SGD trained classifier, which shows poor calibration with a large concentration of extreme confidence values (**Left**); the baseline is a standard, non-private logistic regression model trained by SGD, which is much better calibrated (**Right**).

## 2   RELATED WORK

**Differentially Private Deep Learning.** DP-SGD (Song et al., 2013; Abadi et al., 2016) is a popular algorithm for training deep learning models with DP. Recent works have shown that fine-tuning high-quality pre-trained models with DP-SGD results in good downstream performance (Tramer & Boneh, 2021; Li et al., 2022b; De et al., 2022; Li et al., 2022a). Existing works have studied how ensuring differential privacy through mechanisms such as DP-SGD leads to tradeoffs with other properties, such as accuracy (Feldman, 2020) and fairness (Bagdasaryan et al., 2019; Tran et al., 2021; Sanyal et al., 2022; Esipova et al., 2022) (measured by the disparity in accuracies across groups). Our miscalibration findings are closely related to the above privacy-fairness tradeoff that has already received substantial attention. For example, per-example gradient clipping is shown to exacerbate accuracy disparity (Tran et al., 2021; Esipova et al., 2022). Some fairness notions also require calibrated predictions such as calibration over demographic groups (Pleiss et al., 2017; Liu et al., 2019a) or a rich class of structured "identifiable" subpopulations (Hébert-Johnson et al., 2018; Kim et al., 2019). Our work expands the understanding of tradeoffs between privacy and other aspects of trustworthiness by characterizing privacy-calibration tradeoffs.

**Calibration.** Calibrated probability estimates match the true empirical frequencies of an outcome, and calibration is often used to evaluate the quality of uncertainty estimates provided by ML models. Recent works have observed that highly-accurate models that leverage pre-training are often well-calibrated (Hendrycks et al., 2019; Desai & Durrett, 2020; Minderer et al., 2021; Kadavath et al., 2022). However, we find that even pre-trained models are poorly calibrated when they are fine-tuned using DP-SGD. Our work is not the first to study calibration under learning with DP, but we provide

a more comprehensive characterization of privacy-calibration tradeoffs and solutions that improve this tradeoff which are both simpler and more effective. Luo et al. (2020) studied private calibration for out-of-domain settings, but did not study whether DP-SGD causes miscalibration in-domain. Angelopoulos et al. (2021) modified split conformal prediction to be privacy-preserving, but they only studied vision models and their private models have substantial performance decrease compared to non-private ones. They also did not study the miscalibration of private models and the causes of the privacy-calibration tradeoff. Knolle et al. (2021) studied miscalibration, but only on MNIST and a small pneumonia dataset. Our work provides a more comprehensive characterization across more realistic datasets, and our comparisons show that our recalibration approach is consistently more effective. Closer to our work, the work by Bu et al. (2021) identified that DP-SGD produces miscalibrated models on CIFAR-10, SNLI, and MNIST. As a solution, they suggested an alternative clipping scheme that empirically reduces the expected calibration error (ECE). Our work differs in three ways: our experimental results cover harder tasks and control for confounders such as model accuracy and regularization; we study transfer learning settings that are closer to the state-of-the-art setup in differentially private learning and find substantially worse ECE gaps (e.g. they identify a 43% relative increase in ECE on CIFAR-10, while we find nearly 400% on Food101); we compare our simple recalibration procedure to their method and find that DP-TS is substantially more effective at reducing ECE.

## 3 PROBLEM STATEMENT AND METHODS

Our main goal is to build classifiers that are both accurate and calibrated under differential privacy. We begin by defining core preliminary concepts.

### 3.1 DIFFERENTIAL PRIVACY

Differential privacy is a formal privacy guarantee for a randomized algorithm which intuitively ensures that no adversary has a high probability of identifying whether a record was included in a dataset based on the output of the algorithm. Throughout our work, we will study models trained with approximate-DP / $(\epsilon, \delta)$-DP algorithms.

**Definition 3.1.** (Approximate-DP (Dwork et al., 2006)). The randomized algorithm $\mathcal{M} : \mathcal{X} \to \mathcal{Y}$ is $(\epsilon, \delta)$-DP if for all neighboring datasets $X, X' \in \mathcal{X}$ that differ on a single element and all measurable $Y \subset \mathcal{Y}, \mathbb{P}(\mathcal{M}(X) \in Y) \leq \exp(\epsilon)\mathbb{P}(\mathcal{M}(X') \in Y) + \delta$.

### 3.2 DIFFERENTIALLY PRIVATE STOCHASTIC GRADIENT DESCENT

The standard approach to train neural networks with DP is using the differentially private stochastic gradient descent (DP-SGD) (Abadi et al., 2016) algorithm. The algorithm operates by privatizing each gradient update via combining per-example gradient clipping and Gaussian noise injection.

Formally, one step of DP-SGD to update $\theta$ with a batch of samples $\mathcal{B}_t$ is defined as

$$\theta^{(t+1)} = \theta^{(t)} - \eta_t \left\{ \frac{1}{B} \sum_{i \in \mathcal{B}_t} \text{clip}_C \left( \nabla \mathcal{L}_i \left( \theta^{(t)} \right) \right) + \xi \right\}, \tag{1}$$

where $\eta_t$ is the learning rate at step $t$, $\mathcal{L}\left(\theta^{(t)}\right)$ is the learning objective, $\text{clip}_C\left(\nabla \mathcal{L}_i\left(\theta^{(t)}\right)\right)$ clips the gradient using $\text{clip}_C\left(\nabla \mathcal{L}_i\left(\theta^{(t)}\right)\right) = \nabla \mathcal{L}_i\left(\theta^{(t)}\right) \cdot \min\left(1, C/\|\nabla \mathcal{L}_i\left(\theta^{(t)}\right)\|_2\right)$ and $\xi$ is Gaussian noise defined as $\xi \sim \mathcal{N}\left(0, C^2 \frac{\sigma^2}{B^2} I_p\right)$ with the standard deviation $\sigma$ as the noise multiplier returned by accounting and the expected batch size $B$. Each step of DP-SGD is approximate-DP, and the final model satisfies approximate-DP with privacy leakage parameters that can be computed with privacy loss composition theorems (Abadi et al., 2016; Mironov, 2017; Wang et al., 2019b; Dong et al., 2019; Gopi et al., 2021).

### 3.3 CALIBRATION

A probabilistic forecast is said to be *calibrated* if the forecast has accuracy $p$ on the set of all examples with confidence $p$. Specifically, given a multi-class classification problem where we want to predict a categorical variable $Y$ based on the observation $X$, we say that a probabilistic classifier $h_\theta$ parameterized by $\theta$ over $C$ classes satisfies *canonical calibration* if for each $p$ in the simplex $\Delta^{C-1}$

and every label $y$, $P(Y = y \mid h_\theta(X) = p) = p_y$ holds.[1] Intuitively, a calibrated model should give predictions that can truthfully reflect the predictive uncertainty, e.g., among the samples to which a calibrated classifier gives a confidence 0.1 for class $k$, 10% of the samples actually belong to class $k$.

The canonical calibration property can be difficult to verify in practice when the number of classes is large (Guo et al., 2017). Because of this, we will consider a simpler top-label calibration criterion in this work. In this relaxation, we consider calibration over only the highest probability class. More formally, we say that a classifier $h_\theta$ is calibrated if

$$\forall p^* \in [0, 1], P\left(Y \in \arg\max p \mid \max h_\theta(X) = p^*\right) = p^*, \tag{2}$$

where $p^*$ is the true predictive uncertainty. With the same definition of $p^*$, we will quantify the degree to which a classifier is calibrated through the expected calibration error (ECE), defined by

$$\mathbb{E}\left[\| p^* - \mathbb{E}\left[Y \in \arg\max h_\theta(X) \mid \max h_\theta(X) = p^*\right]\|\right].$$

In practice, we estimate ECE by first partitioning the confidence scores into $M$ bins $B_1, \ldots, B_M$ before calculating the empirical estimate of ECE as

$$\text{ECE} = \sum_{m=1}^{M} \frac{|B_m|}{n} \left|\text{acc}\left(B_m\right) - \text{conf}\left(B_m\right)\right|, \tag{3}$$

where $\text{acc}\left(B_m\right) = \frac{1}{|B_m|} \sum_{i \in B_m} \mathbf{1}\left(y_i = \arg\max h_\theta(\mathbf{x}_i)\right)$, $\text{conf}\left(B_m\right) = \frac{1}{|B_m|} \sum_{i \in B_m} h_\theta(\mathbf{x}_i)$ and $\{(\mathbf{x}_i, y_i)\}_{i=1}^{n}$ are a set of n i.i.d. samples that follow a distribution $P(X, Y)$. When appropriate, we will also study fine-grained miscalibration errors through the histogram of $\text{conf}(B_m)$ (the confidence histogram) and plot $\text{acc}(B_m)$ against $\text{conf}(B_m)$ (the reliability diagram).

### 3.4 RECALIBRATION

When models are miscalibrated, *post-hoc recalibration* methods are often used to reduce calibration errors. These methods alleviate miscalibration by adjusting the log-probability scores generated by the probabilistic prediction model. More formally, consider a score-based classifier that produces probabilistic forecasts via $\text{softmax}(h_\theta(\mathbf{x}))$. We can adjust the calibration of this classifier by learning a $g_\phi$ that adjusts the log probabilities and produces a better calibrated forecast $\text{softmax}(g_\phi \circ h_\theta(\mathbf{x}))$.

Typically, the re-calibration function $g_\phi$ is learned by minimizing a proper scoring rule on a separate validation/recalibration set $X_{\text{recal}}$ with the following optimization problem

$$\min_\phi \mathbb{E}[\ell(\text{softmax}(g_\phi \circ h_\theta(\mathbf{x})), y)]. \tag{4}$$

Specific examples of this type of post hoc recalibration technique include temperature scaling (Guo et al., 2017), Platt scaling (Platt et al., 1999), and isotonic regression (Zadrozny & Elkan, 2002). These methods differ in their choice of $g_\phi$. In this work, we will consider the temperature scaling ($g_\phi(\mathbf{x}) = \mathbf{x}/T$) and the Platt scaling ($g_\phi(\mathbf{x}) = \mathbf{W}\mathbf{x} + \mathbf{b}$) with the choice of log loss for $\ell$.

### 3.5 DIFFERENTIALLY PRIVATE RECALIBRATION

As observed in our motivating example (Fig. 1) and later experimental results (Fig. 2), DP training (with DP-SGD and DP-Adam) tends to produce miscalibrated models. Motivated by the success of recalibration methods such as Platt scaling and temperature scaling in the non-private setting (Guo et al., 2017), we study how these methods can be adapted to build well-calibrated classifiers with DP guarantees.

Our proposed approach is very simple and consists of three steps, shown in Algorithm 1. We first split the training set into a model training part ($X_{\text{train}}$) and a validation/recalibration part ($X_{\text{recal}}$). We then train the model using DP-SGD on $X_{\text{train}}$, followed by a recalibration step using DP-SGD on $X_{\text{recal}}$. Depending on the choice of $g_\phi$, we will refer to this algorithm as either DP temperature scaling (DP-TS) or DP Platt scaling (DP-PS).

---

[1]We slightly abuse the notation of $X$ and $Y$.

---

**Algorithm 1:** Differentially Private Recalibration

---

**Input:** $X = \{(\mathbf{x_1}, y_1), ..., (\mathbf{x_n}, y_n)\}$, validation ratio $\alpha$.
**Initial**: Parameters of models $h_\theta$, recalibrator $g_\phi$.

1. $X_{\text{train}}, X_{\text{recal}} = \text{RandomSplit}(X, \alpha)$

2. Train $h_\theta(\mathbf{x})$ using DP-SGD to optimize $\min_\theta \mathbb{E}\left[\ell\left(\text{softmax}\left(h_\theta(\mathbf{x})\right), y\right)\right]$ with $X_{\text{train}}$

3. Train $g_\phi$ using DP-SGD to optimize $\min_\phi \mathbb{E}\left[\ell\left(\text{softmax}\left(g_\phi \circ h_\theta(\mathbf{x})\right), y\right)\right]$ with $X_{\text{recal}}$

**Output:** $g_\phi \circ h_\theta(\cdot)$

---

The use of sample splitting for recalibration makes privacy accounting simple. To achieve a target $(\epsilon, \delta)$-DP guarantee after recalibration, we can simply run both stages with DP-SGD parameters that achieve $(\epsilon, \delta)$-DP (Prop. A.1). While sample splitting does reduce the number of samples available for model training, using 90% of the dataset for $X_{\text{train}}$ results in a minor utility cost for the model training step in practice.

## 4 EXPERIMENTAL RESULTS

We study three different experimental settings. We first consider **in-domain** evaluations, where we evaluate calibration errors on the same domain that they are trained on. Results show that using pre-trained models does not address miscalibration issues in-domain. We then evaluate the same models above in **out-of-domain** settings, showing that both miscalibration and effectiveness of our recalibration methods carry over to the out-of-domain setting. Finally, we perform careful **ablations** to isolate and understand the causes of in-domain miscalibration. In each case, we will show that DP-SGD leads to high miscalibration, and DP recalibration substantially reduces calibration errors.

**Models.** Our goal is to evaluate calibration errors for state-of-the-art private models. Because of this, our models are based on transfer learning from a pre-trained model. For the text datasets, we fine-tune RoBERTa-base using the procedure in Li et al. (2022b), and for vision datasets, we perform linear probe of ViT and ResNet-50 features, following Tramer & Boneh (2021).

**Datasets.** Following prior work (Li et al., 2022b), we train on MNLI, QNLI, QQP, SST-2 (Wang et al., 2019a) for the text classification tasks, and perform OOD evaluations on common transfer targets such as Scitail (Khot et al., 2018), HANS (McCoy et al., 2019), RTE, WNLI, and MRPC (Wang et al., 2019a).[2] For the vision tasks, we focus on the in-domain setting and evaluate on a subset of the transfer tasks in Kornblith et al. (2019) with at least 50k examples.

**Methods.** As baselines, we train the above models using non-private SGD (NON-PRIVATE), standard DP-SGD (DP), global clipping (Bu et al., 2021) (GLOBAL CLIPPING), and differentially private stochastic gradient Langevin dynamics (Knolle et al., 2021) (DP-SGLD). The last two methods are included to evaluate our simple recalibration approaches against existing methods which are reported to improve calibration.

For our recalibration methods, we run the private recalibration method over the in-domain recalibration set $X_{\text{recal}}$ in Sec. 3.5 using private temperature scaling (DP-TS) (Guo et al., 2017) and Platt scaling (DP-PS) (Platt et al., 1999; Guo et al., 2017). We also include a non-private baseline that combines differentially private model training with non-private temperature scaling (DP+NON-PRIVATE-TS) as a way to quantify privacy costs in the post-hoc recalibration step. Further implementation details and default hyper-parameters for DP training are in Tab. 6 in Appendix B.

### 4.1 IN-DOMAIN CALIBRATION

We now conduct in-depth experiments across multiple datasets and domains to study miscalibration (Tab. 1, 2). We train differentially private models using pre-trained backbones, and find that their accuracies match previously reported high performance (Tramer & Boneh, 2021; Li et al., 2022b; De et al., 2022).

However, we find that these same models have substantially higher calibration errors. For example, the linear probe for Food101 in Fig. 2 has private accuracy within 7% of the non-private counterpart,

---

[2]To match the label space between MNLI and the OOD tasks, we merge "contradiction" and "neutral" labels into a single "not-contradiction" label.

| Category | Model | CIFAR-10 | | SUN397 | | Food101 | |
|---|---|---|---|---|---|---|---|
| | | Accuracy | ECE | Accuracy | ECE | Accuracy | ECE |
| Baseline | DP | 0.7951 | 0.0903 | 0.6844 | 0.1302 | 0.7582 | 0.154 |
| | DP-SGLD | 0.7122 | 0.1331 | 0.6062 | 0.1952 | 0.6476 | 0.2416 |
| | Global Clipping | 0.7712 | 0.0804 | 0.6215 | 0.1125 | 0.7451 | 0.1017 |
| Recalibration | DP-PS | 0.789 | **0.012** | 0.674 | 0.104 | 0.7543 | 0.0554 |
| | DP-TS | 0.789 | 0.0221 | 0.674 | **0.0763** | 0.7543 | **0.0540** |
| Non-private | DP+Non-private-TS | 0.789 | 0.0222 | 0.674 | 0.0764 | 0.7543 | 0.0539 |
| | Non-private | 0.83 | 0.0794 | 0.7044 | 0.1062 | 0.8245 | 0.0349 |

Table 1: The image classification performance ($\epsilon = 8$) of different models before and after recalibration. Results for $\epsilon = 3$ are in Appendix B.3

| Category | Model | MNLI | | QNLI | | QQP | | SST-2 | |
|---|---|---|---|---|---|---|---|---|---|
| | | Accuracy | ECE | Accuracy | ECE | Accuracy | ECE | Accuracy | ECE |
| Baseline | DP | 0.8281 | 0.166 | 0.8503 | 0.149 | 0.8685 | 0.13 | 0.8922 | 0.105 |
| | DP-SGLD | 0.7188 | 0.2625 | 0.7787 | 0.2138 | 0.7917 | 0.2009 | 0.82 | 0.1742 |
| | Global Clipping | 0.8236 | 0.1667 | 0.8502 | 0.1491 | 0.8685 | 0.1296 | 0.8922 | 0.1047 |
| Recalibration | DP-PS | 0.826 | **0.0487** | 0.8464 | **0.0305** | 0.8659 | 0.0672 | 0.8842 | **0.0201** |
| | DP-TS | 0.826 | 0.0849 | 0.8464 | 0.0915 | 0.8659 | **0.0635** | 0.8842 | 0.0665 |
| Non-private | DP+Non-private-TS | 0.826 | 0.0849 | 0.8464 | 0.0915 | 0.8659 | 0.0635 | 0.8842 | 0.0665 |
| | Non-private | 0.8642 | 0.0699 | 0.914 | 0.028 | 0.9042 | 0.0891 | 0.9323 | 0.0425 |

Table 2: The text classification performance ($\epsilon = 8$) before and after recalibration.

but the ECE is more than $4\times$ that of the non-private counterpart. In the language case, we see similar results on QNLI with a ~6% decrease in accuracy but a ~4.3$\times$ increase in ECE. The overall trend of miscalibration is clear across datasets and modalities (Fig. 2).

**DP recalibration.** We now turn our attention to recalibration algorithms and see whether DP-TS and DP-PS can address in-domain miscalibration. We find that DP-TS and DP-PS perform well consistently over all datasets and on both modalities with marginal accuracy drops (Tab.1 and Tab.2). In many cases, the differentially private variants of recalibration work nearly as well as their non-private counterparts. The ECE values for the private DP-TS and non-private baseline of DP+Non-private-TS are generally close across all the datasets.

We note that both DP-TS and DP-PS perform consistently well, with an average relative (in-domain) ECE reduction of 0.58. Despite being simple, the two methods never underperform Global Clipping and DP-SGLD in terms of ECE, and can have very close or even higher accuracies despite the added cost of sample splitting.

**Qualitative analysis.** Examining the reliability diagram before and after DP-TS, we see two clear phenomena. First, the model confidence distribution under DP-SGD is highly polarized (Fig. 3, first two panels) with nearly all examples receiving confidences of 1.0. Next, we see that after DP-TS, this confidence distribution is adjusted to cover a much broader range of confidence values. In the case of SUN397, after recalibration, we see almost perfect agreement between the model confidences and actual accuracies.

### 4.2 OUT-OF-DOMAIN CALIBRATION

We complement our in-domain experiments with out-of-domain evaluations. To do this, we evaluate the zero-shot transfer performance of models trained over MNLI, QNLI (Tab. 3) and QQP (Tab. 4).

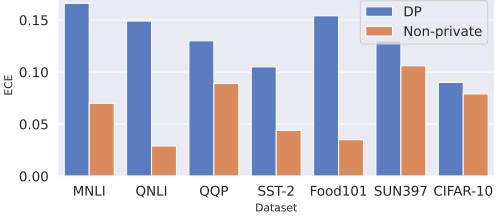

Figure 2: DP trained models display consistently higher ECE than their non-private counterparts.

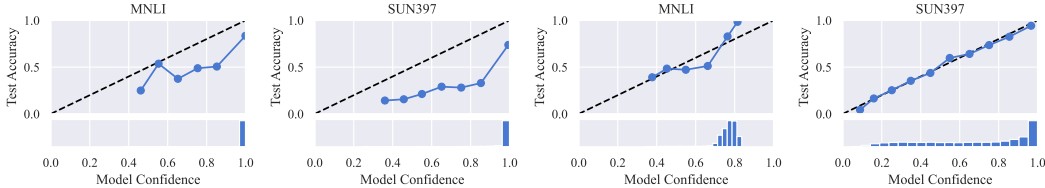

Figure 3: Reliability diagram and confidence histogram before (**Left**) and after (**Right**) recalibration using DP-TS. Recalibration parameters are learned on the validation set $X_{\text{recal}}$ of MNLI and SUN397.

| Dataset | Category | Model | Hans | | Scitail | | RTE | | WNLI | |
|---|---|---|---|---|---|---|---|---|---|---|
| | | | Accuracy | ECE | Accuracy | ECE | Accuracy | ECE | Accuracy | ECE |
| MNLI | Baseline | DP | 0.5195 | 0.4786 | 0.7761 | 0.2172 | 0.7437 | 0.2541 | 0.4507 | 0.5492 |
| | | DP-SGLD | 0.4996 | 0.4995 | 0.7515 | 0.233 | 0.6498 | 0.3169 | 0.4507 | 0.5491 |
| | | Global Clipping | 0.5221 | 0.4747 | 0.7845 | 0.2051 | 0.7076 | 0.2737 | 0.4366 | 0.5632 |
| | Recalibration | DP-PS | 0.5237 | **0.348** | 0.7707 | **0.1089** | 0.7220 | **0.1516** | 0.4366 | **0.4416** |
| | | DP-TS | 0.5237 | 0.3544 | 0.7707 | 0.1168 | 0.7220 | 0.1593 | 0.4366 | 0.4495 |
| | Non-private | DP+Non-private-TS | 0.5237 | 0.3544 | 0.7707 | 0.1168 | 0.7220 | 0.1593 | 0.4366 | 0.4495 |
| | | Non-private | 0.668 | 0.2687 | 0.7853 | 0.1348 | 0.7906 | 0.1518 | 0.507 | 0.4677 |
| QNLI | Baseline | DP | 0.5046 | 0.4932 | 0.729 | 0.2666 | 0.5657 | 0.4407 | 0.4724 | 0.5215 |
| | | DP-SGLD | 0.5 | 0.4986 | 0.7209 | 0.2723 | 0.5668 | 0.4266 | 0.4225 | 0.5738 |
| | | Global Clipping | 0.5025 | 0.4971 | 0.7293 | 0.2684 | 0.5199 | 0.4761 | 0.4789 | 0.52 |
| | Recalibration | DP-PS | 0.5002 | **0.3244** | 0.7377 | **0.0832** | 0.5632 | **0.2578** | 0.4648 | **0.3464** |
| | | DP-TS | 0.5002 | 0.385 | 0.7377 | 0.1353 | 0.5632 | 0.3121 | 0.4648 | 0.404 |
| | Non-private | DP+Non-private-TS | 0.5002 | 0.385 | 0.7377 | 0.1353 | 0.5632 | 0.3121 | 0.4648 | 0.404 |
| | | Non-private | 0.538 | 0.1969 | 0.7454 | 0.0690 | 0.5199 | 0.3036 | 0.5493 | 0.2438 |

Table 3: The **zero-shot transfer** NLI performance ($\epsilon = 8$) across multiple OOD test datasets.

Our findings are consistent with the in-domain evaluations. Differentially private training generally results in high ECE, while DP-TS and DP-PS generally improve calibration. The gaps out-of-domain are substantially smaller than the in-domain case, as all methods are of low accuracy and miscalibrated out of domain. However, the general ranking of miscalibration methods, and the observation that DP-TS and DP-PS lead to private models with calibration errors on-par to non-private models is unchanged.

### 4.3 ANALYSES AND ABLATION STUDIES

Finally, we carefully study two questions to better understand the miscalibration of private learners: What component of DP-SGD leads to miscalibration? What are other confounders such as accuracy or regularization effects that lead to miscalibration?

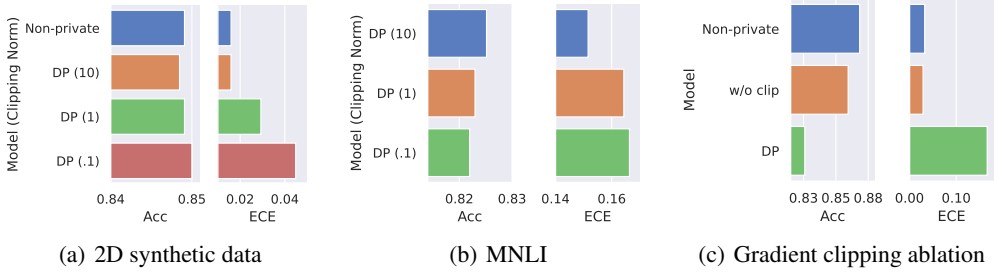

(a) 2D synthetic data      (b) MNLI      (c) Gradient clipping ablation

Figure 4: Per-example gradient clipping ($\epsilon = 8$) causes large ECE errors in (a) logistic regression on non-separable 2D synthetic data, and (b) fine-tuning RoBERTa on MNLI. (c) Performing only gradient noising leads to high accuracy and low ECE.

**Ablation on per-example gradient clipping and noise injection.**   DP-SGD involves per-example gradient clipping and noise injection. To better understand which component contributes more to miscalibration, we perform experiments to isolate the effect of each individual component.

On 2D synthetic data (example given in Fig. 1), Fig. 4(a) shows that fixing the overall privacy guarantee ($\epsilon$) and increasing the clipping threshold from DP (0.1) to DP (1) and further to DP (10)

| Dataset | Category | Model | MRPC | |
| --- | --- | --- | --- | --- |
| | | | **Accuracy** | **ECE** |
| QQP | Baseline | DP | 0.7475 | 0.252 |
| | | DP-SGLD | 0.6936 | 0.2979 |
| | | Global Clipping | 0.7475 | 0.252 |
| | Recalibration | DP-PS | 0.7426 | **0.1252** |
| | | DP-TS | 0.7426 | 0.1796 |
| | Non-private | DP+Non-private-TS | 0.7426 | 0.1796 |
| | | Non-private | 0.7255 | 0.2635 |

Table 4: The **zero-shot transfer** paraphrase performance ($\epsilon = 8$) from QQP to MRPC.

affect the accuracy only marginally but substantially improve calibration. Repeating this ablation with RoBERTa fine-tuning on MNLI (Fig. 4(b)) confirms that increasing the clipping threshold (slightly) decreases ECE but does not substantially impact model accuracy. Finally, Fig. 4(c) shows that completely removing clipping and training with only noisy gradient descent dramatically reduces ECE (and increases accuracy). These results suggest that intensive clipping exacerbates miscalibration (even under a fixed privacy guarantee).

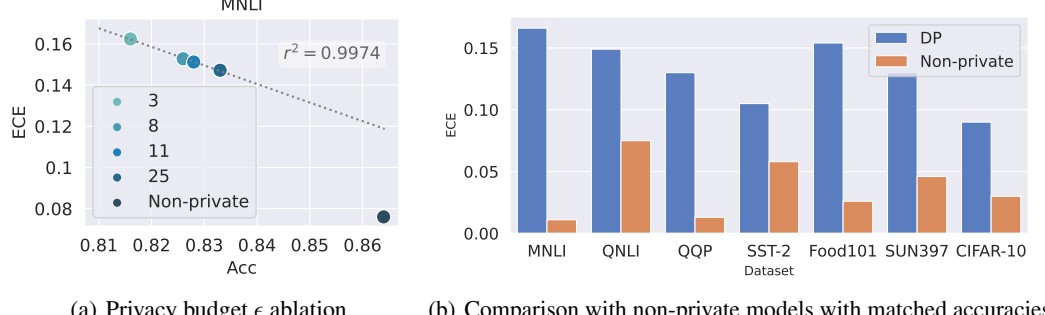

(a) Privacy budget $\epsilon$ ablation   (b) Comparison with non-private models with matched accuracies

Figure 5: (a) MNLI performance under varying **privacy budgets** $\epsilon$. (b) **Controlling for accuracy** by early stopping non-private models to match the DP models does not substantially affect differences in ECE. The accuracy differences are within $1\%$.

| Method | MNLI | | Scitail | | QNLI | |
| --- | --- | --- | --- | --- | --- | --- |
| | **Accuracy** | **ECE** | **Accuracy** | **ECE** | **Accuracy** | **ECE** |
| DP | 0.8281 | 0.166 | 0.7761 | 0.2172 | 0.5058 | 0.4942 |
| Non-private | 0.8642 | 0.0699 | 0.7853 | 0.1348 | 0.5050 | 0.4426 |
| $\ell_2$ (1e-4) | 0.8664 | 0.0347 | 0.7876 | 0.0822 | 0.5058 | 0.4874 |
| $\ell_2$ (1e-3) | 0.8672 | 0.0349 | **0.7891** | **0.0816** | 0.5056 | **0.4862** |
| $\ell_2$ (1e-2) | 0.8620 | **0.0326** | 0.7845 | 0.0845 | 0.5059 | 0.4870 |
| $\ell_2$ (1e-1) | **0.8684** | 0.1874 | 0.786 | 0.0835 | **0.5059** | 0.4872 |
| dropout (0.1) | **0.8684** | 0.1874 | **0.786** | **0.0835** | **0.5059** | 0.4872 |
| dropout (0.2) | 0.8601 | **0.046** | 0.7722 | 0.1076 | 0.5058 | 0.487 |
| dropout (0.3) | 0.8380 | 0.0629 | 0.7423 | 0.1523 | 0.5050 | **0.486** |
| early stopping (2) | 0.8423 | **0.0288** | 0.7806 | **0.0662** | 0.5050 | **0.4818** |
| early stopping (4) | 0.8572 | 0.0299 | 0.78 | 0.094 | 0.5058 | 0.486 |
| early stopping (6) | 0.8623 | 0.0355 | 0.7837 | 0.0811 | 0.5056 | 0.486 |
| early stopping (8) | **0.8684** | 0.1874 | **0.786** | 0.0835 | **0.5059** | 0.4872 |

Table 5: Comparison with non-private models trained using common **regularizers**, i.e. $\ell_2$ (weight decay factor), dropout (probability) and early stopping (total training epochs). Models are trained on MNLI and evaluated over MNLI, Scitail and QNLI.

**Controlling for accuracy and regularization.**   Accuracy and calibration are generally positively correlated (Minderer et al., 2021; Carrell et al., 2022). This poses a question: Does the miscalibration of DP models arise due to their suboptimal accuracy? We find evidence against this in two different experiments.

In the first experiment, we vary $\epsilon$ for fine-tuning RoBERTa with DP on MNLI. This results in several models situated on a linear ECE-accuracy tradeoff curve (Fig. 5(a)). Intuitively, extrapolating this

curve helps us identify the anticipated ECE for a DP trained model with a given accuracy. Fig. 5(a) shows that when compared to these private models, the non-private model has substantially lower ECE than would be expected by extrapolating this tradeoff alone. This suggests that private learning experiences a qualitatively different ECE-accuracy tradeoff than standard learning.

In the second experiment, we controlled the in-domain accuracy of non-private models to match their private counterparts by early-stopping the non-private models to be within 1% of the DP model accuracy. Fig. 5(b) shows that the ECE gap between the private and non-private models persists even when controlling for accuracy.

More generally, we find that regularization methods such as early stopping impact the ECE-accuracy tradeoff qualitatively differently than DP-SGD. Our results in Tab. 5 show that most other regularizers such as early-stopping lead to an accuracy-ECE tradeoff, in which highly regularized models are less accurate but better calibrated. This is not the case for DP training, where the resulting models are

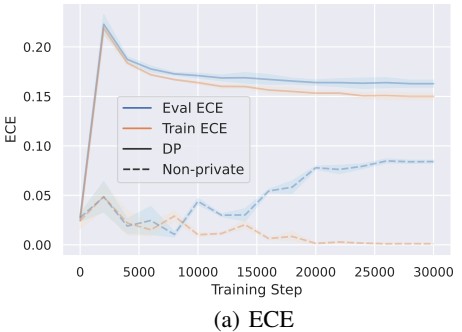
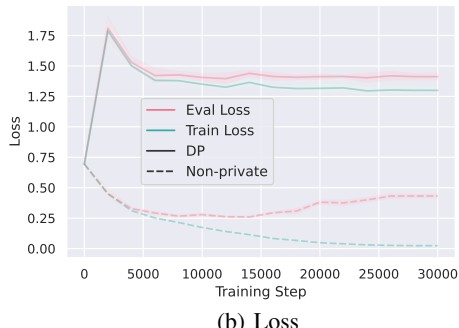

(a) ECE

(b) Loss

Figure 6: **DP-SGD training ($\epsilon = 8$) makes train and eval ECE close but both of them are large.** The training dynamics of (a) ECE and (b) Loss on both QNLI training and evaluation sets.

both of lower accuracy and less calibrated relative to their non-private counterparts. These findings suggest that calibration errors in private and non-private settings may be caused by different reasons - the miscalibration of private models may not be due to the regularization effects of DP-SGD.

**DP training leads to similarly high train and test ECE.** Learning algorithms which satisfy tight DP guarantees are known to generalize well, meaning that the train (empirical) and test (population) losses of a DP trained model should be similar (Dwork et al., 2015; Bassily et al., 2016). In a controlled experiment, we fine-tune RoBERTa on QNLI with DP-SGD ($\epsilon = 8$) and observe that the train-test gaps for both ECE and loss are smaller for DP models than the non-private ones (Fig. 6). Yet, for DP trained models, both the train and test ECEs are high compared to the non-private model. Interestingly, these observations with DP trained models are very different from what's seen in miscalibration analyses of non-private models. For instance, Carrell et al. (2022) showed that non-private models tend to be calibrated on the training set but can be miscalibrated on the test set due to overfitting (large *calibration generalization gap*). Our results show that DP trained models have a small calibration generalization gap, but are miscalibrated on both the training and test sets.

## 5 CONCLUDING REMARKS

In this work, we study the calibration of ML models trained with DP-SGD. We quantify the miscalibration of DP-SGD trained models and verify that they exist even using state-of-the-art pre-trained backbones. While the calibration errors are substantial and consistent, we show that adapting existing post-hoc calibration methods is highly effective for DP-SGD models. We believe it is an open question whether it is possible to leverage the generalization guarantees of DP-SGD to naturally obtain similarly well-calibrated models without the use of sample-splitting and recalibration.

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

# A  PRIVACY ANALYSIS FOR INDEPENDENT RELEASES WITH A PARTITION OF DATA

Our post-processing calibration setup requires splitting the original (private) training data into two disjoint splits where one of which is used solely for training and the other solely for post hoc recalibration. Given that both the training and post hoc recalibration algorithms are DP, it is natural to ask what is the overall privacy spending of the joint release. While one can essentially resort to any "off-the-shelf" privacy composition theorem, we note that in our setup the splits of data used in the two algorithms are disjoint, and thus a tighter characterization of privacy leakage is possible. The following is common knowledge, and we only include the proof for completeness.

**Proposition A.1.** *Let $M_1 : \mathcal{X}_1 \to \mathcal{Y}$ and $M_2 : \mathcal{X}_2 \times \mathcal{Y} \to \mathcal{Z}$ be $(\epsilon, \delta)$-DP algorithms consuming independent random bits operating on disjoint splits of the dataset. Then, the algorithm $M : \mathcal{X} \to \mathcal{Y} \times \mathcal{Z}$ defined by*

$$M(X) = (y, z), \quad y = M_1(X_1), \quad z = M_2(X_2, y),$$

*where $(X_1, X_2)$ is a partition of $X$ determined through some procedure independent on $X$, is also $(\epsilon, \delta)$-DP.*

*Proof.* Let $X$ and $X'$ be neighboring datasets. Suppose that the first component in both partitions is the same, i.e., $X = (X_1, X_2)$, and $X' = (X_1, X_2')$, where $X_2$ and $X_2'$ are neighboring. Then, $M$ is $(\epsilon, \delta)$-DP directly follows from that $M_2$ is $(\epsilon, \delta)$-DP.

The more subtle case is when the second component in both partitions is the same. Specifically, suppose that $X = (X_1, X_2)$, and $X' = (X_1', X_2)$, where $X_1$ and $X_1'$ are neighboring. Let $R$ denote the random variable that controls only the randomness of $M_2$, i.e., conditioned a draw of $R = r$, $M_2$ is a deterministic function. With slight abuse of notation, we denote this deterministic function by $M_2(r)$. Let $O = \cup_{o_1 \in O_1} \{o_1\} \times O_2(o_1) \subset \mathcal{Y} \times \mathcal{Z}$ be a subset of the codomain. Define the following shorthand for the preimage of $M_2$ conditioned on $R = r$

$$M_2(r)^{-1}(X_2, S) = \{y \in \mathcal{Y} \mid M_2(r)(X_2, y) \in S\}.$$

Then, we have

$$\Pr\left(M(X) \in O \mid R = r\right) = \sum_{o_1 \in O_1} \Pr\left(M_1(X_1) = o_1\right) \Pr\left(M_2(X_2, o_1) \in O_2(o_1) \mid R = r\right)$$

$$= \sum_{o_1 \in O_1} \Pr\left(M_1(X_1) = o_1\right) \mathbb{1}\left[M_2(r)(X_2, o_1) \in O_2(o_1)\right]$$

$$= \sum_{o_1 \in O_1} \Pr\left(M_1(X_1) = o_1, \, M_1(X_1) \in M_2(r)^{-1}(X_2, O_2(o_1))\right)$$

$$= \Pr\left(M_1(X_1) \in \cup_{o_1 \in O_1}\left(\{o_1\} \cap M_2(r)^{-1}(X_2, O_2(o_1))\right)\right)$$

$$\leq e^\epsilon \Pr\left(M_1(X_1') \in \cup_{o_1 \in O_1}\left(\{o_1\} \cap M_2(r)^{-1}(X_2, O_2(o_1))\right)\right) + \delta$$

$$= e^\epsilon \Pr\left(M(X') \in O \mid R = r\right) + \delta.$$

Since the above holds for all draws of $R$, we conclude that $\Pr\left(M(X) \in O\right) \leq e^\epsilon \Pr\left(M(X') \in O\right) + \delta$ for all neighboring $X$ and $X'$ which differ only in their first components. This concludes the proof. $\square$

# B  EXTENDED EXPERIMENTAL DETAILS AND RESULTS

## B.1  SETTINGS FOR SYNTHETIC EXPERIMENTS

For **synthetic** experiments, we generate two-dimensional mixture Gaussian data of size $10k$. The distance between the centers of two class data shifts by a constant, which is set to be $2 * 1.5$. We use logistic regression to do the binary classification. We set the amount of data points from each class as 5k and batch size as $4k$. We include the results with different maximum gradient norm $C \in \{0.1, 0.5, 1\}$ of DP training.

| Dataset | CIFAR-10 | SUN397 | Food101 | MNLI | QNLI | QQP | SST-2 |
|---|---|---|---|---|---|---|---|
| Learning rate | 2e-3 | 1e-2 | 1e-4 | 5e-4 | 1e-3 | 5e-4 | 1e-3 |
| Batch size | 32 | 32 | 32 | 6,000 | 2,000 | 6,000 | 1,000 |
| LR decay | False | False | False | True | True | True | True |
| Epochs | 10 | 10 | 10 | 18 | 6 | 18 | 3 |
| Weight decay | 1e-4 | 1e-4 | 1e-4 | 0 | 0 | 0 | 0 |
| Clipping norm | 1.0 | 1.0 | 1.0 | 0.1 | 0.1 | 0.1 | 0.1 |
| Privacy budget $\epsilon$ | 3, 8 | 3, 8 | 3, 8 | 8 | 8 | 8 | 8 |
| Validation ratio | | | | 0.1 | | | |
| Noise scale | | | calculated numerically so that a DP budget of $(\epsilon, \delta)$ is spent after E epochs | | | | |

Table 6: Default hyperparameter of DP finetuning over different datasets for reproducibility. Batch size is based on a unit batch size 20 with different amount of gradient accumulation steps. We use the validation ratio, the proportion of validation set, to split the training set for tuning recalibration methods.

## B.2 IMPLEMENTATION DETAILS

We use pre-trained checkpoints and trainers from Huggingface library (Wolf et al., 2020) for NLP experiments. We do linear probe for CV experiments using ResNet50 for CIFAR-10, ViT for SUN397 and Food101. We use the modified Opacus privacy engine (Yousefpour et al., 2021) from (Li et al., 2022b), which computes per-example gradients for transformers. We compare DP training with popular regularizers used for finetuning like $\ell_2$, dropout and early stopping over NLP datasets. $\ell_2$ is the weight decay rate $\{1e-1, 1e-2, 1e-3, 1e-4\}$ during optimization. We apply dropout to both hidden and attention layers of transformers, which takes the value in $\{0.1, 0.2, 0.3, 0.4\}$. We do early stopping by setting the maximum amount of training epochs to be smaller, i.e. values in $\{2, 4, 6, 8\}$. The default hyper-parameters for $\ell_2$, dropout, early stopping are $1e-1$, $0.1$, $8$ respectively so some of the results in Tab.5 are reused.

For recalibration training, we use a fixed amount of epochs without hyper-parameter tuning to avoid privacy leakage of validation sets. We initialize the temperature parameter in DP-TS as 1.0 and train 100 epochs for all the tasks except Food1001 (which uses 30 epochs) using DP-SGD with a 0.1 learning rate, 10 maximum gradient clipping norm, and a linearly decayed learning rate scheduler. We adapt multiclass extensions for Platt scaling by considering higher-dimensional parameters (Guo et al., 2017).

For baselines, we grid search the maximum norm bound $Z \in \{100, 500, 1000\}$ and epochs over $\{6, 8, 18\}$ for global clipping (Bu et al., 2021); we use pre-noise scale 0.046, temperature $\tau = 6.08$, exponential learning rate decay with learning rate 0.005 and decay factor 0.028 as suggested by Knolle et al. (2021).

## B.3 ADDITIONAL IMAGE CLASSIFICATION RESULTS

In Tab. 7, we give additional results when we have a smaller privacy budget $\epsilon = 3$. We see consistent results that DP fine-tuning gives poor calibration performance while DP-TS and/or DP-PS can recalibrate the classifiers effectively.

| Category | Model | CIFAR-10 | | SUN397 | | Food101 | |
|---|---|---|---|---|---|---|---|
| | | **Accuracy** | **ECE** | **Accuracy** | **ECE** | **Accuracy** | **ECE** |
| Baseline | DP | 0.7912 | 0.0916 | 0.6751 | 0.2806 | 0.7097 | 0.2464 |
| | DP-SGLD | 0.6953 | 0.1595 | 0.562 | 0.3295 | 0.6217 | 0.2834 |
| | Global Clipping | 0.7659 | 0.0782 | 0.6345 | 0.285 | 0.6853 | 0.2276 |
| Recalibration | DP-PS | 0.7823 | **0.0109** | 0.6694 | 0.2826 | 0.7084 | 0.0626 |
| | DP-TS | 0.7823 | 0.0217 | 0.6694 | **0.0183** | 0.7084 | **0.0601** |
| Non-private | DP+Non-private-TS | 0.7823 | 0.0218 | 0.6694 | 0.019 | 0.7084 | 0.0598 |
| | Non-private | 0.83 | 0.0794 | 0.7044 | 0.1062 | 0.8245 | 0.0349 |

Table 7: The image classification performance ($\epsilon = 3$) of different models before and after recalibration across datasets.

### B.4 ADDITIONAL ABLATION STUDIES

**Label noise injection.** All of the datasets we consider have labels that are designed to be unambiguous, and the Bayes optimal predictor would produce a confidence histogram that is concentrated at 1.0. In this case, we might wonder whether the polarized confidence histograms observed in Fig. 3 are an artifact for datasets with unambiguous labels.

To understand this, we intentionally inject label noise into MNLI and study how this changes the behavior of DP-SGD and non-private learning algorithms. Specifically, we uniformly corrupt training labels - by selecting a uniform random class with probability $p \in \{0.6, 0.8\}$. We compare DP-SGD trained models and non-private models with $0.2$ dropout regularization. The confidence histograms in Fig. 7 clearly demonstrate that differentially private models result in 100% confidence, *even when the Bayes optimal classifier can be at most 60% confident*. This shows that DP-SGD trained model's miscalibration behavior that results in near 100% confidence is not driven by a dataset's label distribution and this behavior is likely to be even worse on tasks with inherent label uncertainty.

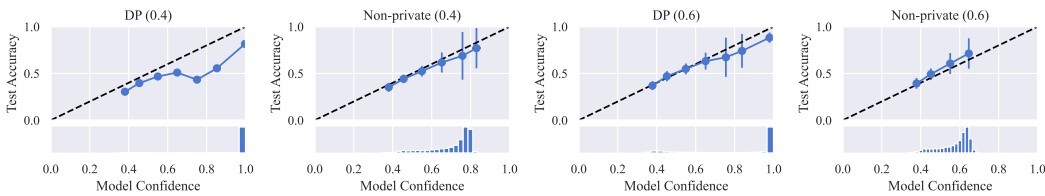

Figure 7: Reliability diagram and confidence histogram for **label noise** settings with different models (corruption rates) trained on MNLI. For comparison, non-private models are included.

## C REMARKS ON CORRELATIONS BETWEEN ACCURACY AND CALIBRATION

In general, the correlations between accuracy and calibration are not clearly understood even for non-private learners as many factors can impact calibration such as architecture, regularization, optimization, data distribution, overparameterization, etc. Below we include some notable empirical findings. Convolutional networks like ResNets and DenseNets can be miscalibrated (Guo et al., 2017). However, Minderer et al. (2021) show that modern models like ViT (Dosovitskiy et al., 2020) are better calibrated compared to past models; modern neural networks tend to have a strong positive correlation between calibration and classification error; model architectures matter greatly in calibration properties. Using pre-training can improve model uncertainty and calibration (Hendrycks et al., 2019; Desai & Durrett, 2020; Minderer et al., 2021; Kadavath et al., 2022). Regularizations like gradient noise injection can promote stability and distributional generalization so good calibration over the training set can transfer to the test set (Kulynych et al., 2022). Carrell et al. (2022) empirically shows that popular models with small generalization gaps will have small test calibration errors.

Realizing the above observations, it is possible that the per-example gradient clipping and gradient noise injection in DP-SGD can contribute to both accuracy and calibration in different ways. Therefore, we carefully control the accuracy and regularization when conducting analyses and drawing conclusions (Tab. 5, Fig. 5(a) and 5(b), Fig. 7). However, even with the confounding controls above, DP-SGD trained models are still miscalibrated. In other words, the reason for the finding that private learners are much more miscalibrated than non-private counterparts is less likely to be the unambiguous labels in datasets, accuracy discrepancy or regularization effects of DP-SGD but more likely to be the per-example gradient clipping operation.

