# OpenReview forum: "A Closer Look at the Calibration of Differentially Private Learners"
_ICLR.cc/2023/Conference — Submitted to ICLR 2023_

### Official Review · Reviewer_s6au · 2022-10-17

**Confidence:** 3
**Correctness:** 4
**Technical Novelty And Significance:** 2
**Empirical Novelty And Significance:** 3
**Recommendation:** 6

**Clarity, Quality, Novelty And Reproducibility:**

The description of Algorithm 1 is very general. There is no detailed explanation of how DP-SGD is applied in both step 2 and step 3 of Algorithm 1. The authors should give more details.

The idea of calibration is not quite clear. It is not clear to me the underlying motivation on why the calibration is a big issue we need to consider in algorithm design. What is its connection to accuracy? There is not enough background on calibration.

In the abstract, the paper says their analysis identifies per-example gradient clipping as a major issue of miscalibration. However, I do not find such an analysis in the paper.

Below eq (1): $l_i$ should be $L_i$

**Strength And Weaknesses:**

**strength**

The proposed method is simple and general. It provides a framework which includes several post-hoc recalibration techniques. There are many experimental results, which show the proposed method works well in practice.

**weakness**

There is no theoretical analysis of the proposed method. There is no theoretical guarantee and therefore it is not clear how the method would behave in general.

The idea of extending recalibration to differentially private recalibration seems to be natural. It is not clear whether the extension is novel enough.

**Summary Of The Paper:**

This paper studies the calibration of differentially private learners based on stochastic gradient descent. The paper first observes the miscalibration  is due to the per-example gradient clipping. Then, the paper provides differentially private recalibration to reduce  calibration errors. The basic idea is to divide the training dataset into two parts: one part is used to train a classifier, while the other is used to train a recalibration function. Extensive experimental results are reported to show the effectiveness of the differentially recalibration method.

**Summary Of The Review:**

The paper presents a simple and general method for recalibration in a differentially private setting. The paper conducts many experiments to show the effectiveness of the method. The proposed algorithm is not presented in detail and there is no theoretical analysis.

---

> ### Author Response · Authors · 2022-11-14
> **Response to Reviewer s6au**
>
> Thank you for the constructive feedback and comments. Please let us know if there is any concern we can address further.
>
> > There is no theoretical analysis of the proposed method.
>
> Please see the **General Response** and **Proposition A.1**.
>
> > The novelty of the idea of extending recalibration to differentially private recalibration
>
> We argue that the novelty of our DP recalibration are from three aspects:
>
> **Effectiveness**: there is no existing effective approach to do recalibration while preserving privacy guarantees for state-of-the-art DP learners. Across 7 tasks, DP-TS and DP-PS with DP-SGD produce an average 3.1-fold reduction in the in-domain expected calibration error and only incur at most a minor percent drop in accuracy. In comparison, as in our comprehensive experiments, those previous work like global clipping and DP-SGLD are shown to be ineffective in reducing calibration errors.
>
> **Reusing existing recalibration algorithms with simplicity**: TS and PS are effective recalibration methods that have stood against the test of time and reusing them is intuitive and well-motivated. As in Algorithm 1 and supported by proposition A.1, mitigating the miscalibration using DP-TS and DP-PS is substantially simple so that they can be much more likely to be adopted compared with complicated counterparts like global clipping and DP-SGLD. For example, the reason why (non-private) TS is widely adopted is partly because it is very simple yet effective.
>
> **Guarantee**: to the best of our knowledge, it is new for proposition A.1 to guarantee that splitting the training set and running the two stages using the same privacy budget does not affect the overall privacy budget.
>
> > Detailed explanations of how DP-SGD is applied in both step 2 and step 3 of Algorithm 1
>
> The concrete procedure of DP-SGD is included in Sec. 3.2, which involves per-example gradient clipping and gradient noise injection for both step 2 and step3.
> DP-SGD is an optimization algorithm for learning the parameters $\theta$ and $\phi$ for step2 and step3, respectively. Please note that we have specified the training objective, model/parameter, inputs in Algorithm 1, Sec. 3.4 and Sec. 3.5. In Sec. 3.4, with the choice of log loss for $\ell$, we show that in we use $\left(g_\phi(\mathbf{x})=\mathbf{x} / T\right)$ for DP-TS and $\left(g_\phi(\mathbf{x})=\mathbf{W} \mathbf{x}+\mathbf{b}\right)$ for DP-PS.
> > Underlying motivation on why the calibration is a big issue we need to consider in algorithm design. What is its connection to accuracy? There is not enough background on calibration.
>
> Please see the **General Response** and we have revised the paper to include more background about the motivations and significance of calibration in Sec.1. The connection of calibration to accuracy is added in Appendix C.
>
> > The analysis that identifies per-example gradient clipping as a major issue of miscalibration.
>
> The analysis is supported by experimental results:
> Fig. 4(a) and 4(b) show that decreasing the maximum gradient clipping norm $C$ leads to higher ECE while the accuracy remains merely the same (note that the smaller $C$ it is, the more aggressive clipping is);
> Fig. 4(c) shows that if we completely remove the clipping, the accuracy and ECE will become very similar to the non-private model, which verifies that gradient noise injection alone does not lead to miscalibration while clipping is a significant cause for miscalibration.
> Since the procedure of DP-SGD only involves per-example gradient clipping and gradient noise injection, we study those two steps separately to control confounders in the above experiments.
>
> > Below eq (1): li should be Li
>
> Thanks for pointing out the consistency problem and we’ve fixed it in our updated draft.

---

### Official Review · Reviewer_zkmx · 2022-10-20

**Confidence:** 4
**Correctness:** 4
**Technical Novelty And Significance:** 2
**Empirical Novelty And Significance:** 2
**Recommendation:** 6

**Clarity, Quality, Novelty And Reproducibility:**

Writing Quality:
1. I think the paper is a bit vague at different places, and the notations are not well-defined. In Section 3.3 while defining canonical calibration, what is $h_{\theta}$? What is the probability over? Is $Y$ the true label or is it the prediction of $h$? Also, is Equation 2 like an analogue of $\ell_{\infty}$ error? In the same subsection, what is $\hat{p}_i$?
2. Might want to motivate in a paragraph why calibration is an important problem to study even if we have good accuracy. Understanding the motivation of this is important, and I don't think the authors do a great job at conveying that.
3. The DP-PS and DP-TS are unexplained in this draft. I know you use DP-SGD here, but giving some kind of an analysis or more detailed descriptions of the two would make it much more meaningful.
4. Might want to give a clear intuition of what calibration actually means or entails. Might make sense to explain the formal definitions a bit in words to show what it means to have bad calibration even if we have good accuracy.

Novelty:
There is novelty in the sense that there are previous works that address this issue under DP in settings different from the one in this paper.

**Strength And Weaknesses:**

Strengths:
1. The paper successfully depicts the issue of miscalibrations in DP classifiers (trained via DP-SGD). All the experiments indicate the DP counterparts having more miscalibration than their respective non-private counterparts.
2. This paper also establishes its second goal of providing a DP method to lower the miscalibration errors significantly. These methods involve the use of DP-SGD again. The experimental results indicate very low miscalibration errors that are similar to those from the non-private methods.

Weaknesses:
1. I have some concerns about the writing quality. Please, see the section regarding clarity of the submission.
2. I'm also concerned about the significance of the problem being solved here. I'm curious as to why this problem under DP has not been studied much before this. In other words, why is reducing miscalibrations that important a problem to solve, especially under DP?
3. The algorithmic contributions don't seem to be that interesting, unfortunately -- just optimising a calibration function using DP-SGD. The techniques aren't novel, and simply involve blackbox uses of DP-SGD. Is the goal of this paper to bring the issue of miscalibrations in DP classifiers to people's attention?

**Summary Of The Paper:**

This paper studies the calibration of state-of-the-art classifiers trained using differentially private stochastic gradient descent (DP-SGD). It shows that there are miscalibration issues in these DP classifiers even when their accuracy matches that of their non-private counterparts. The likely cause of that is conjectured to be the repeated per-example gradient clipping in DP-SGD. The other contribution is addressing these miscalibrations using DP versions of temperature scaling (TS) and Platt scaling (PS), which are almost as good as their non-private counterparts. This work is mostly empirical in nature, and uses multiple datasets and models, with applications in text-processing and computer vision, and has experiments to evaluate both in-domain and out-of-domain calibrations.

**Summary Of The Review:**

I am not convinced about the overall quality of this paper. The contributions don't seem that significant to me, unfortunately, especially given the lack of motivation of the problem in this draft and the not-so-novel technical ideas. The writing quality wasn't that great either, but that doesn't matter much to me as far as the quality of the work is concerned -- my feedback there is just to help the authors improve their manuscript. I do acknowledge that the authors seem to have achieved the claims of this paper. All this said, I will be open to changes in my final score based on the authors' responses though.

Update: Based on the responses, I have bumped the score.

---

> ### Author Response · Authors · 2022-11-14
> **Response to Reviewer zkmx [1/2]**
>
> Thank you for the constructive feedback and comments. Please let us know if there is any concern we can address further.
>
> > Writing clarity
>
> **In Section 3.3, definition of $h_\theta$? What is the probability over? Is Y the true label or is it the prediction of h? Also, is Equation 2 like an analogue of $\ell_\infty$ error? In the same subsection, what is $\hat{p}_i$?**
>
> Thanks for pointing these out and we’ve fixed those notations for consistency:
> - Please note that we define the model parameters in Sec. 3.2, so $h_\theta$ is a probabilistic classifier parameterized by parameters $\theta$.
> - Y is the prediction of h.
> - Eq. (2) is the standard definition of canonical calibration based on top-label calibration criterion rather than a $\ell_\infty$ error for optimization. It means that we convert the confidences into the class label with maximum confidence.
> - $\hat{p}_i$ is the predictive confidence $h_\theta(x_i)$ on sample $x_i$.
>
> **The descriptions of DP-TS and DP-PS.**
>
> Please note that we introduce the abbreviation in Sec.1 and details in the Sec 3.4 and Algorithm 1: they are post-hoc recalibration methods for reducing calibration errors. They alleviate miscalibration by adjusting the log-probability scores generated by the probabilistic prediction model. More formally, consider a score-based classifier that produces probabilistic forecasts via softmax $\left(h_\theta(\mathbf{x})\right)$. We can adjust the calibration of this classifier by learning a $g_\phi$ that adjusts the log probabilities and produces a better calibrated forecast softmax $\left(g_\phi \circ h_\theta(\mathbf{x})\right)$.
>
> Typically, the re-calibration function $g_\phi$ is learned by minimizing a proper scoring rule on a separate validation/recalibration set $X_{\text {recal}}$ with the following optimization problem
> $ \min _\phi \mathbb{E}\left[\ell\left(\operatorname{softmax}\left(g_\phi \circ h_\theta(\mathbf{x})\right), y\right)\right] .$
> With the choice of log loss for $\ell$, DP-TS and DP-PS differ in their choice of $g_\phi$: DP-TS uses $\left(g_\phi(\mathbf{x})=\mathbf{x} / T\right)$ and the DP-PS uses $\left(g_\phi(\mathbf{x})=\mathbf{W} \mathbf{x}+\mathbf{b}\right)$
>
> **Might want to motivate in a paragraph why calibration is an important problem to study even if we have good accuracy. Understanding the motivation of this is important.**
>
> Please see the **General Response**. Moreover, we have revised the paper to include more background about the motivations and significance of calibration in Sec.1.
>
> **What calibration actually means or entails. Might make sense to explain the formal definitions a bit in words to show what it means to have bad calibration even if we have good accuracy.**
>
> Please see the **General Response** and we have revised the paper to include more background about the motivations and significance of calibration in Sec.1. The remark on the connection of calibration to accuracy is added in Appendix C.
>
> > Why this problem under DP has not been studied much before this? Why is reducing miscalibrations that important a problem to solve, especially under DP?
>
> We note in the Related Work that existing works found that DP could cause miscalibration but they did not comprehensively characterize the privacy-calibration trade-off as we do: (1) our experimental results cover harder tasks and control for confounders such as model accuracy and regularization; we study transfer learning settings that are closer to the state-of-the-art setup in differentially private learning and find substantially worse ECE gaps (e.g. they identify a 43% relative increase in ECE on CIFAR-10, while we find nearly 400% on Food101); we compare our simple recalibration procedure to their method and find that DP-TS is substantially more effective at reducing ECE.
>
> Miscalibration under DP is a significant problem and trustworthy aspects are inner-connected as discussed in the **General Response**. Use cases that need protecting privacy often involve consequential scenarios where users want to access the uncertainty of models.

---

> > ### Author Response · Authors · 2022-11-14
> > **Response to Reviewer zkmx [2/2]**
> >
> > > The novelty of DP recalibration
> >
> > We argue that the novelty of our DP recalibration is from three aspects:
> >
> > **Effectiveness**: there is no existing effective approach to do recalibration while preserving privacy guarantees for state-of-the-art DP learners. Across 7 tasks, DP-TS and DP-PS with DP-SGD produce an average 3.1-fold reduction in the in-domain expected calibration error and only incur at most a minor percent drop in accuracy. In comparison, as in our comprehensive experiments, those previous work like global clipping and DP-SGLD are shown to be ineffective in reducing calibration errors.
> >
> > **Reusing existing recalibration algorithms with simplicity**: TS and PS are effective recalibration methods that have stood against the test of time and reusing them is intuitive and well-motivated. As in Algorithm 1 and supported by proposition A.1, mitigating the miscalibration using DP-TS and DP-PS is substantially simple so that they can be much more likely to be adopted compared with complicated counterparts like global clipping and DP-SGLD. For example, the reason why (non-private) TS is widely adopted is partly because it is very simple yet effective.
> >
> > **Guarantee**: to the best of our knowledge, it is new for proposition A.1 to guarantee that splitting the training set and running the two stages using the same privacy budget does not affect the overall privacy budget.
> >
> > > Is the goal of this paper to bring the issue of miscalibrations in DP classifiers to people's attention?
> >
> > Please note that our goal is more about (a) showing empirically that miscalibration of DP classifiers (privacy-calibration trade-off) is a noteworthy research problem by comprehensively characterizing the privacy-calibration trade-off and identifying the cause of miscalibration and (b) proposing a very practical and effective solution to mitigate miscalibration.
> > Therefore, bringing the issue of miscalibrations in DP classifiers to people's attention can be one of our goals.

---

> ### Author Response · Authors · 2022-11-29
> **A Gentle Reminder of Response**
>
> Dear Reviewer zkmx,
>
> We thank you for your invaluable feedback to enhance our submission. As the discussion period draws to a close, we would like to kindly remind you to review our responses and revisions to ensure that all your concerns have been addressed.
>
> We have worked hard to carefully address each point with detailed explanations and results, and the draft has been revised accordingly. We sincerely value your input and would be immensely grateful if you could confirm that our responses and revisions have adequately addressed your concerns.
>
> Sincerely,
>
> Authors

---

### Official Review · Reviewer_9fSP · 2022-10-22

**Confidence:** 3
**Correctness:** 2
**Technical Novelty And Significance:** 1
**Empirical Novelty And Significance:** 4
**Recommendation:** 6

**Clarity, Quality, Novelty And Reproducibility:**

The problem and the experiments are clearly presented. The comparison with previous work is also discussed.

**Strength And Weaknesses:**

The paper provides extensive experimental evidence about the impact of differential privacy to the calibration property of classifiers. I find that the provided empirical observations are interesting and may raise some nice theoretical questions too. More to that, the paper is nicely written. The only issue is that the paper does not have any theoretical contribution.

A potential question is whether variants of DP-SGD or other training methods would also cause such miscalibration phenomena. Is gradient clipping the only obstacle towards private training that is well-calibrated?

**Summary Of The Paper:**

This paper studies the trade-off between differential privacy and classifier calibration. In particular, the main observation is that classifiers trained with the standard DP-SGD algorithm can be highly miscalibrated. To this end, the authors propose a natural and intuitive solution to fix this issue using differentially private recalibration. First, the algorithm runs DP-SGD in order to obtain the standard classification model with a subset of the data and then runs DP-SGD again in order to find a post-hoc recalibration function using the remaining data. Finally, it outputs the composition of the two functions.


**Summary Of The Review:**

In short, even if there is no theoretical contribution, I believe that the paper is beyond the acceptance threshold, since its message can potentially raise some theoretical questions. The experimental observations are interesting.

---

> ### Author Response · Authors · 2022-11-14
> **Response to Reviewer 9fSP**
>
> Thank you for the constructive feedback and comments. Please let us know if there is any concern we can address further.
>
> > Theoretical developments
>
> Please see the **General Response** and **Proposition A.1**.
>
> > Whether variants of DP-SGD or other training methods would also cause such miscalibration phenomena
>
> Please note that the DP-SGD variants global clipping and DP-SGLD are included in our experiments as baselines, showing that those methods that were claimed to address the miscalibration issue are ineffective in our settings: models trained using them get similar or higher ECE than that trained using DP-SGD.
>
> > Is gradient clipping the only obstacle towards private training that is well-calibrated?
>
> As we claim in the paper, we identify per-example gradient clipping in DP-SGD as a major cause of miscalibration as opposed to the only obstacle. The finding is supported by experimental results:
> Fig. 4(a) and 4(b) show that decreasing the maximum gradient clipping norm $C$ leads to higher ECE while the accuracy remains merely the same (note that the smaller $C$ it is, the more aggressive clipping is);
> Fig. 4(c) shows that if we completely remove the clipping, the accuracy and ECE will become very similar to the non-private model, which verifies that gradient noise injection alone does not lead to miscalibration while clipping is a significant cause for miscalibration.
>
> Since the procedure of DP-SGD only involves per-example gradient clipping and gradient noise injection, we study those two steps separately to control confounders in the above experiments.
>
> And as per-example gradient clipping is necessary to preserve the privacy guarantee in DP-SGD, it is significant to identify its tangible effect on causing miscalibration.

---

### Official Review · Reviewer_ckRn · 2022-10-24

**Confidence:** 3
**Correctness:** 3
**Technical Novelty And Significance:** 2
**Empirical Novelty And Significance:** 2
**Recommendation:** 5

**Clarity, Quality, Novelty And Reproducibility:**

The paper is clearly written, the quality of the empirical work seems fine but there is a lack of adequate theoretical developments. The algorithm may have some elements of novelty, and the results look reproducible.



**Strength And Weaknesses:**

Strengths:


Authors have a good grasp of some of the challenges of differentially private computations in the context of the paper. The paper is generally well-written. The empirical context is well motivated.


Weakness:


This paper is primarily on empirical evidence for a recalibration idea. It is not clear to me that the recalibration preserves differential privacy, although there is claim in the paper about this. Perhaps additional discussion on this in an eventually revised manuscript is in order. There does not seem to be adequate theoretical justification about when, why and how the standard DP-SGD method fails, or when, why and how the proposed recalibration will work, or when it will fail.


**Summary Of The Paper:**

Authors  identify the per-example gradient clipping in differentially private stochastic gradient descent (DP-SGD) as a major cause of miscalibration, and argue that existing baselines for improving private calibration only provide small improvements in calibration error while occasionally causing large degradation in accuracy. Authors propose using post-processing calibration.



**Summary Of The Review:**

The main emphasis of this work is empirical evidence. More detailed analysis is needed.

---

> ### Author Response · Authors · 2022-11-14
> **Response to Reviewer ckRn**
>
>
> Thank you for the constructive feedback and comments. Please let us know if there is any concern we can address further.
>
> > How the recalibration preserves differential privacy
>
> Please note that we include how and why the recalibration step preserves DP in section 3.5: The use of sample splitting for recalibration makes privacy accounting simple. Assuming the data order is set beforehand (not depending on the algorithm and their randomness), running two independent (eps, delta)-DP algorithms on two disjoint splits of the data is still considered (eps, delta)-DP. (Prop. A.1). The claim is supported by Proposition A.1 in Appendix A.
>
> > Theoretical developments
>
> Please see the **General Response** and **Proposition A.1**.

---

> ### Author Response · Authors · 2022-11-29
> **A Gentle Reminder of Response**
>
> Dear Reviewer ckRn,
>
> We thank you for your invaluable feedback to enhance our submission. As the discussion period draws to a close, we would like to kindly remind you to review our responses and revisions to ensure that all your concerns have been addressed.
>
> We have worked hard to carefully address each point with detailed explanations and results, and the draft has been revised accordingly. We sincerely value your input and would be immensely grateful if you could confirm that our responses and revisions have adequately addressed your concerns.
>
> Sincerely,
>
> Authors

---

### Author Response · Authors · 2022-11-14
**General Response [1/3]**

We thank all the reviewers for their valuable comments and constructive suggestions. Reviewers agreed that the paper was well-motivated (ckRn, 9fSP), novel and interesting (9fSP, zkmx), well-written (ckRn, 9fSP) and our claims are well-supported and correct (zkmx, s6au). We have revised the paper accordingly and the revised parts are highlighted with red color.

Reviewers had some shared concerns that we detail below and further address in the individual author's reply. Please let us know if there are any additional results or textual clarifications that we can provide.

----

> The definition, motivations and significance of calibration, especially under DP. (zkmx, s6au)

**Definition of calibration.**

A formal definition is in Sec 3.3. Intuitively, a calibrated model should give predictions that can truthfully reflect the predictive uncertainty, e.g., among the samples to which a calibrated classifier gives a confidence 0.1 for class k, 10% of the samples actually belong to class k.

**The connection between calibration and accuracy.**

We include more remarks in Appendix C. In general, the correlation between accuracy and calibration error is still an open research question as discussed above [12]. As shown in Appendix B.2 of [12], modern neural networks tend to have a strong positive correlation between calibration and classification error. However, given the calibration definition above, a classifier with high accuracy does not necessarily have good calibration. For example, a highly accurate but miscalibrated classifier can always output polarized confidence scores so the top-class confidence is always above 0.9. This is corroborated both qualitatively (Fig. 3) and quantitatively (Tab. 1 and Tab.2) in our experiments. And a highly accurate but poorly calibrated model may not be able to be deployed safely as it can cause high risks to society [5].

**The significance and motivations of calibration.**

We argue that calibration is an important fundamental property of ML models that have been discussed in numerous ML tutorials and textbooks such as [1, 2, 3] and miscalibration is widespread for deep learning [4]. Calibration is significant for deploying models in safety-critical scenarios like healthcare and autonomous driving where accessing model uncertainty is important [5]. One main reason is that we need to know when the predictions may go wrong so we can use techniques like selective prediction for risk control in those use cases. Another reason is that calibrated confidence estimates are also important for providing model explainability to establish trustworthiness with the user [6].

[1] Probabilistic Machine Learning: An Introduction. https://probml.github.io/pml-book/book1.html

[2] Sklearn Probability calibration. https://scikit-learn.org/stable/modules/calibration.html

[3] The Well-alibrated Bayesian. http://fitelson.org/seminar/dawid.pdf

[4] Guo, C., Pleiss, G., Sun, Y. and Weinberger, K.Q., 2017, July. On calibration of modern neural networks. ICML 2017.

[5] Van Calster, B., McLernon, D.J., Van Smeden, M., Wynants, L. and Steyerberg, E.W., 2019. Calibration: the Achilles heel of predictive analytics. BMC medicine, 17(1), pp.1-7.

[6] Cosmides, L. and Tooby, J., 1996. Are humans good intuitive statisticians after all? Rethinking some conclusions from the literature on judgment under uncertainty. cognition, 58(1), pp.1-73.

---

> ### Author Response · Authors · 2022-11-14
> **General Response [2/3]**
>
>
> **The significance and motivations for calibration under DP**
>
> We argue that the calibration problem of DP-SGD trained models is significant and well-motivated because of the following reasons.
>
> (a) In practice, deploying models reliably to safety-critical scenarios that need privacy protection usually requires promoting other key aspects of trustworthiness such as calibration [7] and fairness [8]. And these tradeoffs have already received substantial attention and study in the context of fairness-privacy tradeoffs [8,15] to prevent disparate societal and economic impacts on the involved individuals. This issue is especially pressing for recent applications of pre-trained models to areas like healthcare and laws [9].
>
> (b) There are recent findings that modern pre-trained models can be better calibrated [10, 11, 12]. Moreover, based on pre-trained backbones, private models can also be boosted with accuracies approaching or matching their non-private counterparts [13]. So an open question is does the increased accuracy of private learners with pre-training lead to improved calibration? Our work is the first to use extensive experiments and careful confounding control to arrive at a negative answer.
>
> (c) There are many works studying the privacy-fairness trade-off [8, 14, 15] while the privacy-calibration trade-off is poorly understood and not comprehensively identified for state-of-the-art private learners using pre-trained models.
>
> (d) Our findings can be surprising to practitioners studying DP since DP (with reasonably tight bounds / small parameters) is expected to guarantee generalization and good calibration in theory due to its stability guarantee (Theorem C.8 in [16]). But it can be understood in our newly added experiments that DP-SGD makes both training and test ECE high even though the ECE generalization gap is small (Fig. 6). The story is not the same for non-private models that tend to be calibrated on the training set but can be miscalibrated on the test set due to overfitting [17].
>
> [7] Knolle, M., Ziller, A., Usynin, D., Braren, R., Makowski, M.R., Rueckert, D. and Kaissis, G., 2021. Differentially private training of neural networks with Langevin dynamics for calibrated predictive uncertainty. arXiv preprint arXiv:2107.04296.
>
> [8] Bagdasaryan, E., Poursaeed, O. and Shmatikov, V., 2019. Differential privacy has disparate impact on model accuracy. NeurIPS 2019.
>
> [9] Bommasani, R., Hudson, D.A., Adeli, E., Altman, R., Arora, S., von Arx, S., Bernstein, M.S., Bohg, J., Bosselut, A., Brunskill, E. and Brynjolfsson, E., 2021. On the opportunities and risks of foundation models. arXiv preprint arXiv:2108.07258.
>
> [10] Hendrycks, D., Lee, K. and Mazeika, M., 2019, May. Using pre-training can improve model robustness and uncertainty. ICML 2019.
>
> [11] Tran, D., Liu, J., Dusenberry, M.W., Phan, D., Collier, M., Ren, J., Han, K., Wang, Z., Mariet, Z., Hu, H. and Band, N., 2022. Plex: Towards reliability using pretrained large model extensions. arXiv preprint arXiv:2207.07411.
>
> [12] Minderer, M., Djolonga, J., Romijnders, R., Hubis, F., Zhai, X., Houlsby, N., Tran, D. and Lucic, M., 2021. Revisiting the calibration of modern neural networks. NeurIPS 2021.
>
> [13] Li, X., Tramer, F., Liang, P. and Hashimoto, T., 2021. Large language models can be strong differentially private learners. ICLR 2021.
>
> [14] Feldman, V., 2020, June. Does learning require memorization? a short tale about a long tail. SIGACT Symposium on Theory of Computing (pp. 954-959).
>
> [15] Sanyal, A., Hu, Y. and Yang, F., 2022. How unfair is private learning?. UAI 2022.
>
> [16] Kulynych, B., Yang, Y.Y. and Nakkiran, Y.Y.J.B.P., What You See is What You Get: Principled Deep Learning via Distributional Generalization. NeurIPS 2021.
>
> [17] Carrell, A., Mallinar, N., Lucas, J. and Nakkiran, P., 2022. The Calibration Generalization Gap. arXiv preprint arXiv:2210.01964.

---

> > ### Author Response · Authors · 2022-11-14
> > **General Response [3/3]**
> >
> >
> > > The theoretical development. (ckRn, s6au, 9fSP)
> >
> > - Please note that proposition A.1 guarantees the privacy budget spending so it is useful for end-to-end private algorithm designs due to training set splitting. To the best of our knowledge, the guarantee that splitting the training set and running the two stages using the same privacy budget do not affect the overall privacy budget is new.
> >
> > - In general, we do not believe that every publication in an ML conference should include theoretical contributions. And our empirical results are much stronger than previous works: (a) we have identified a much greater relative increase in ECE (nearly 400% on Food101); (b) DP-TS and DP-PS result in an average 3.1-fold reduction in the in-domain ECE across 7 tasks; (c) we empirically verify that private learners have similarly high training and test ECE even if the calibration generalization gap is small, which is unexplored for existing theories. There are also numerous empirical works that do not involve any theory but still made substantial contributions and progress to the field [12, 17, 19, 21]. And oftentimes they inspire many theoretical follow-up works to explain and justify their previous empirical findings ([17]->[13], [12]->[18], [19]->[20], [21]->[22]).
> >
> > - Please note that the connections between calibration and accuracy itself is an open research problem without clear principles and theoretical guarantees [11, 16], as highlighted in **Appendix C**. And we believe our empirical findings about train vs test ECE give a direct target for future theoretical and algorithmic improvements. That’s because it can narrow the scope for what assumptions on data, models and causes for miscalibration should look like in a future theory: empirically, we have comprehensively conducted controlled experiments to identify a much worse calibration error of private learners than previous works and shed light on the causes for it; the newly added experiments show that DP-SGD makes both training and test ECE high even though the ECE generalization gap is small (Fig. 6).
> >
> > [17] Zhang, C., Bengio, S., Hardt, M., Recht, B. and Vinyals, O., 2021. Understanding deep learning (still) requires rethinking generalization. Communications of the ACM, 64(3), pp.107-115.
> >
> > [18] Li, X., Liu, D., Hashimoto, T., Inan, H.A., Kulkarni, J., Lee, Y.T. and Thakurta, A.G., 2022. When Does Differentially Private Learning Not Suffer in High Dimensions?. NeurIPS 2022.
> >
> > [19] Nakkiran, P., Kaplun, G., Bansal, Y., Yang, T., Barak, B. and Sutskever, I., 2021. Deep double descent: Where bigger models and more data hurt. Journal of Statistical Mechanics: Theory and Experiment, 2021(12), p.124003.
> >
> > [20] Nakkiran, P., Venkat, P., Kakade, S. and Ma, T., 2020. Optimal regularization can mitigate double descent. ICLR 2021.
> >
> > [21] Brown, T., Mann, B., Ryder, N., Subbiah, M., Kaplan, J.D., Dhariwal, P., Neelakantan, A., Shyam, P., Sastry, G., Askell, A. and Agarwal, S., 2020. Language models are few-shot learners. NeurIPS 2020.
> >
> > [22] Xie, S.M., Raghunathan, A., Liang, P. and Ma, T., 2021. An explanation of in-context learning as implicit bayesian inference. ICLR 2022.

---

### Author Response · Authors · 2022-11-14
**Revision Summary**

We thank the reviewers for constructive feedback and have updated our draft based on the suggestions:

- We included more background about the definition, intuition, importance and significance of calibration, especially under DP in Sec.1 and Sec. 3.3.
- We showed in Sec. 2 that existing privacy-fairness tradeoffs are significant and widely studied, and they are closely related to the privacy-calibration tradeoff that we would like to characterize.
- We added a remark in Appendix C discussing the correlation between calibration and accuracy. And based on those existing common observations, our finding that DP-SGD leads to significant miscalibration even for modern pre-trained models can be surprising.
- We included additional experimental results for a better understanding of miscalibration in Sec. 4.3. We show that DP-SGD leads to similarly high train and test ECE even though DP guarantees that train and test ECE are close, which is unexpected in previous works.
- We revised and unified the notations in Sec. 3.2 and  Sec 3.3 for consistency.
- We added $r^2$ in Fig. 5(a) showing the substantial differences in accuracy-ECE tradeoffs of private and non-private learners.

The above revised parts are highlighted in red color and please let us know if there are any additional results or textual clarifications that we can provide.

---

### Decision · Program_Chairs · 2023-01-20

**Decision:**

Reject

**Justification For Why Not Higher Score:**

There does not seem to be adequate theoretical justification about when, why and how the standard DP-SGD method fails, or when, why and how the proposed recalibration will work, or when it will fail.
Extending recalibration to differentially private recalibration seems to be natural: it is  unclear whether the extension is novel enough.

**Justification For Why Not Lower Score:**

It is a clearly written paper.
The empirical work is fine.
The algorithm may have some elements of novelty, and the results look reproducible.


**Metareview: Summary, Strengths And Weaknesses:**

The paper studies the calibration of differentially private learners based on stochastic gradient descent.
It first observes that the miscalibration is due to the per-example gradient clipping. Then, the paper provides differentially private recalibration to reduce calibration errors. The basic idea is to divide the training dataset into two parts: one part is used to train a classifier, while the other is used to train a recalibration function (standard or Platt scaling). Extensive experimental results are reported to show the effectiveness of the differentially recalibration method.



**Summary Of Ac-Reviewer Meeting:**

No meeting. The reviewer never interacted on this paper.